# The Role of Preconception Parental Health on Embryo Quality—Preliminary Results of a Prospective Study Using Non-Invasive Preimplantation Genetic Testing for Aneuploidy

**DOI:** 10.3390/biom15091215

**Published:** 2025-08-22

**Authors:** Maja Tomic, Eda Bokal-Vrtacnik, Martin Stimpfel

**Affiliations:** 1Department of Human Reproduction, Division of Obstetrics and Gynaecology, University Medical Centre Ljubljana, 1000 Ljubljana, Slovenia; maja.tomic@kclj.si (M.T.); eda.vrtacnikbokal@kclj.si (E.B.-V.); 2Faculty of Medicine, University of Ljubljana, 1000 Ljubljana, Slovenia

**Keywords:** non-invasive preimplantation genetic testing for aneuploidy, next-generation sequencing, embryo ploidy, male weight, female age, body mass index

## Abstract

In this study, we aimed to correlate embryonic ploidy status studied with non-invasive preimplantation genetic testing for aneuploidy with the basic patient characteristics of the infertile couple to gain insight into the effects of parental physical health on embryo ploidy. We recruited 131 couples, who were stratified into 4 groups based on female age. We gathered general patient characteristics of the couple and determined the female’s hormonal status. We included 316 embryos in our study. Embryos were either transferred in the uterus in a fresh cycle or vitrified for later use. We collected spent embryo culture medium on either day 5 or 6 and performed whole genome amplification before using Next Generation Sequencing. Pregnancy outcomes were noted and cross-referenced with patient characteristics and the embryo’s ploidy status in a retrospective manner. While we have indirectly observed a level of maternal contamination, we nevertheless found a significant correlation between embryo ploidy status and cell free deoxyribonucleic acid concentration in spent embryo culture, as well a correlation between female age and embryo ploidy status. We observed a significant correlation between male body mass index and cell free deoxyribonucleic acid concentration in spent embryo culture medium and between male body mass index and pregnancy outcome. We illustrated a connection between male body mass index and cell free deoxyribonucleic acid, independent of female markers. This is the first study to observe not only female but male parameters in correlation to cell free deoxyribonucleic acid.

## 1. Introduction

Numerous biological parameters affect human fecundity. The effect of female age on female reproduction has been widely researched, with clear evidence indicating a decline in fecundity in women with advancing age [1,2]. Furthermore, female obesity has been shown to affect both the number of retrieved oocytes [3] and embryo quality [4]. In contrast, the role of male biological parameters in reproductive outcomes remains less well-defined and studied [5,6]. Some studies suggest that sperm aneuploidy [7] and DNA fragmentation [8] could be present even in sperm with normal motility and concentration, potentially influencing pregnancy outcomes. Additionally, emerging evidence indicates that male obesity could influence pregnancy outcomes [9,10]. However, limited data is available regarding parameters that affect embryo ploidy. While there is some evidence that embryo ploidy declines with advancing female age [11], few studies have systematically investigated other contributing parameters. One such study, conducted by Fouks et al., examined the influence of sperm-related factors on embryo ploidy and reported no significant correlations [12]. However, a key limitation of their analysis was the exclusion of mosaic embryos, which may have impacted the overall findings.

One of the most widely used methods of determining embryo ploidy is preimplantation genetic testing for aneuploidy (PGT-A), an invasive method using trophectoderm biopsy (TE) to gain access to genetic material [13,14,15]. The identification of cell-free deoxyribonucleic acid (cfDNA) in spent embryo culture medium (SECM) allowed for the evaluation of embryo ploidy without compromising embryonic integrity—the method is called non-invasive preimplantation genetic testing for aneuploidy (niPGT-A) [16,17,18,19]. In theory, testing SECM, as opposed to a limited number of TE cells, should yield results that are more concordant with embryo ploidy [18,20,21,22,23]. Although the origins of cfDNA have not yet been confirmed [19,24], several studies have nevertheless demonstrated niPGT-A to be a promising method in establishing embryo ploidy [18,19,22,25]. Different authors have theorized that cfDNA could, in part, be due to cell apoptosis [24,26,27]. This could explain why studies like Huang et al. [22] in 2019 and Li et al. in 2021 [28] concluded that niPGT-A method was promising, as they froze and re-thawed their embryos, which could in turn promote cell apoptosis and elevate cfDNA concentrations in SECM.

While niPGT-A offers a promising new alternative, it is not yet used in clinical practice. One of the obstacles remains the variable concordance rate between niPGT-A and conventional PGT-A reported in different studies [18,19,22,25,29].

The application of niPGT-A in a pre-clinical setting has predominantly focused on determining embryo ploidy and evaluating its concordance with results obtained via conventional PGT-A. While PGT-A is one of the most widely used methods, it is limited in use by its invasiveness. In practice, this means PGT-A is usually offered to a specific selection of patients. Because niPGT-A is non-invasive, it can be offered to a wide variety of patients. It could also help study embryo ploidy and parental characteristics in relations to it, without ever compromising an embryo’s integrity.

The primary objective of the study was to investigate the correlation between embryonic ploidy status determined with niPGT-A with basic patient characteristics of the infertile couple and pregnancy outcomes.

## 2. Materials and Methods

### 2.1. Study Design

The current prospective study was conducted on 131 couples who received infertility treatment at the Department of Human Reproduction, University Medical Centre (UMC) Ljubljana, Slovenia. All couples signed informed consent forms before being included in the study.

Inclusion criteria for females were age greater than 18 years and treatment received due to a male factor of infertility. Exclusion criteria for women were age greater than 45 years and a known genetic disorder. Inclusion criteria for men were a poor semen sample (teratozoospermia, oligozoospermia, asthenozoospermia, or any combination of the three) that would require the use of Intracytoplasmic Sperm Injection (ICSI). Exclusion criteria for men also included the presence of a known genetic disorder. There was no age restriction for men.

Women underwent routine hormonal screening prior to an assisted reproductive technology (ART) procedure. Couples were stratified in Groups 1 through 4 based on female age.

Women underwent either a short antagonist protocol or long agonist protocol for ovarian stimulation. Both protocols were as described previously [30]. Oocyte and sperm collection and preparation ahead of the ICSI procedure were performed as previously described [30,31].

### 2.2. Embryo Culture and Collection of Samples

After ICSI, the oocytes were cultured all together in Universal IVF Medium (Origio, Måløv, Denmark) overnight. After an oocyte fertilization check, normally fertilized oocytes (those containing two pronuclei) were transferred into a 30 µL droplet of pre-incubated SAGE-1 Step (Origio) culture medium made in a Micro-droplet culture dish (Vitrolife, Västra Frölunda, Sweden). The culture medium droplets were covered with 4 mL of paraffin oil (Origio). Each embryo was cultured in a separate droplet, marked with an individual indicator, for 5 or 6 days, without additional culture medium changes or washing. On days 5–6 embryos were graded as previously described [32]. The best quality embryo was transferred into the uterus. All other suitable embryos were vitrified for transfer at a later date using a Kitazato Vitrification Kit.

After an embryo was removed from the culture for either transfer or vitrification, we collected 20 µL of SECM, which we froze for future analysis at −80 °C. Leftover SECM was discarded.

### 2.3. Result Analysis

Samples of SECM were transferred to Amplexa Genetics (Amplexa Genetics A/S, Odense, Denmark), where they underwent in depth analysis according to the company’s established protocols.

Briefly, cfDNA in the SECM samples was amplified directly in the medium using whole genome amplification (WGA). This was followed by Indexing PCR, during which adapters and barcodes were attached. These procedures were performed under conditions suitable for low-input DNA. Samples were then first pooled based on DNA concentration and then based on the molarity of the library. In the next step, next generation sequencing (NGS) was performed using an Illumina 6000 platform. Interpretation was based on the data output of minimally 500,000 single-end reads per sample, resulting in the resolution of 10 million base pairs (Mbp). This resolution allowed the detection of whole chromosome aneuploidies and segmental aneuploidies that were at least 10 Mbp in size as well as mosaicism.

### 2.4. Statistical Analysis

Statistical analysis was performed using IBM SPSS Statistics 27.0 software (IBM Corporation, Armonk, NY, USA). Kolmogorov–Smirnov and Shapiro–Wilk tests were used to assess the normality of the data distribution. Since the data did not follow a normal distribution, they are presented as medians along with the lower (Q1) and upper (Q3) quartiles. The data presented as medians was analyzed using the Kruskal–Wallis test for independent samples to compare the groups. When the *p*-values were under 0.05, Dunn’s post hoc test was applied for pair-wise comparisons, and the adjusted *p*-values were reported. The correlations between different variables were tested using Spearman’s correlation test. The categorical variables were analyzed using Pearson’s chi-square test or Fisher’s exact test as appropriate, and a z-test was applied for post-hoc analysis. A *p*-value of <0.05 was considered statistically significant throughout the analysis. The G∗Power program, version 3.1.9.7 [33] was used to calculate the statistical power of the analysis and the sample size of embryos needed to detect a statistically significant difference with a power of 0.8 (the calculation was performed specifically for the analysis of the euploid embryo rate).

### 2.5. Female Age Groups

A total of 131 couples undergoing infertility treatment at the UMC Ljubljana and had oocyte retrieval between 1 January 2022 and 4 November 2023 were included in this prospective study. Each couple was included with only one cycle in this study. Couples were included and stratified into groups 1 through 4 based on female age. There were 29 couples in Group 1 with women aged from 18 to 31, 38 couples in Group 2 with women aged between 31 and 35, 33 couples in Group 3 with wo, men aged between 35 and 40, and 31 couples in Group 4 with women aged 40 or more.

Embryonic ploidy status was coded numerically. Number 0 indicated an unknown ploidy status, number 1 marked euploid embryos, number 2 marked aneuploid embryos, number 3 marked mosaic embryos. We used number 4 to indicate unassigned data.

## 3. Results

### 3.1. Correlation of Results Based on Female Age Groups

The general characteristics of each group are shown in Table 1. Overall, there was a significant decline of anti-Müllerian hormone (AMH) between all age groups (*p* = 0.031), but post-hoc analysis revealed a significant decline of AMH only between the youngest and oldest age groups (1 vs. 4 *p* = 0.034. There was no significant difference between the different age groups regarding follicle-stimulating hormone (FSH) levels, luteinizing hormone (LH) levels, prolactin (PRL) levels, thyroid stimulating hormone (TSH) levels, and male or female body mass index (BMI). The total dose of used gonadotrophins significantly increased with aging for all groups (*p* < 0.001), but post-hoc analysis revealed the only significant differences in total dose of used gonadotrophins were between Groups 1 vs. 3 (*p* = 0.004) and Groups 1 vs. 4 (*p* < 0.001), while no difference in other comparisons was found.

In vitro fertilization (IVF) cycle outcomes by age groups are presented in Table 2. There were no significant differences between the age groups and the number of normally fertilized oocytes per number of oocytes injected (*p* = 0.727), immature oocytes (*p* = 0.864), degenerated oocytes per number of retrieved oocytes (*p* = 0.864), and polyploidies per number of retrieved oocytes (*p* = 0.999). There were no significant differences between the different age groups and the number of embryos (*p* = 0.357) or blastocysts per cycle (*p* = 0.138). On the other hand, the blastocyst rate (per embryos cultured until day 5/6) differed significantly between groups (*p* = 0.004). The rate was significantly higher in group 1 versus group 2 (*p* = 0.007), versus group 3 (*p* = 0.020), and versus group 4 (*p* < 0.001). Comparisons among other groups did not demonstrate any significant differences. Moreover, the quality of blastocysts (classified as good, fair, or poor) also appeared to be comparable across the different age groups.

Embryo utilization and pregnancy outcomes based on different age group categories are shown in Table 3. The utilization rate was significantly higher in Group 1 versus Group 2 (*p* = 0.008) and in Group 1 versus Group 4 (*p* = 0.023), while the rate of cryopreserved blastocysts was significantly higher in Group 1 versus Group 4 (*p* = 0.002) and in Group 2 versus Group 4 (*p* = 0.024). There was no significant difference between age group categories when the rate of cycles with at least one blastocyst on day 5/6 and rate of cycles with embryo cryopreservation were compared. Our study had no cases of cycles without embryo transfer (ET), as those couples were excluded from the study. What is more, no significant differences were observed among the age groups in terms of the proportion of pregnancies achieved per fresh ET or per frozen ET. On the other hand, a significant difference was observed in both the cumulative pregnancy rate (*p* = 0.015) and the cumulative delivery rate (*p* = 0.023). Pairwise comparison of individual age groups for cumulative pregnancy rate revealed significant differences between Group 1 versus 3 (*p* = 0.017), Group 1 versus 4 (*p* = 0.004), and Group 2 versus 4 (*p* = 0.038), while other comparisons did not reveal differences. Similarly, the comparison of individual age groups for cumulative delivery rate demonstrated significant differences between Group 1 versus 4 (*p* = 0.016), Group 2 versus 3 (*p* = 0.043) and Group 2 versus 4 (*p* = 0.006), while other comparisons did not reveal any differences.

The collection of samples and cfDNA analysis in different age groups is shown in Table 4. Most of the samples were collected on day 5, and the proportion between the age groups in similar. No SECM was collected on day 7. There was no significant difference between age groups in the amount of cfDNA collected on day 5 or on day 6.

The euploid embryo rate also appeared to differ between age groups (*p* = 0.042). However, post-hoc analysis revealed significant differences only between Group 1 and Group 4 (*p* = 0.008), and between Group 3 and Group 4 (*p* = 0.039). A subsequent post hoc power and sample size analysis revealed that the statistical power of the euploid embryo rate analysis was 0.64, with an effect size of 0.161. A sample size calculation determined (with power 0.8 and with effect size 0.161) that altogether a minimum of 421 embryos should be included into the study to achieve significant differences (*p* < 0.05) in this category, and to be exact (to replicate the current p-value (0.042)), a total of 441 embryos would be necessary. When this number of embryos (421) is converted to the number of couples, considering current number of obtained embryos and embryo development, at least 43 additional couples should be included into the study to attain adequate statistical power. There was no significant difference in the male euploid embryo rate, but the data show the sex of embryos was mostly determined as female (see Table 4). The mosaic embryo rate appeared to differ across age groups (*p* = 0.043), but post-hoc analysis identified significant differences between Groups 2 and 3 (*p* = 0.047) and Groups 3 and 4 (*p* = 0.002), although this finding is likely to lack clinical relevance due to the limited number of included samples/embryos.

### 3.2. Results Using the Spearman’s Rank Correlation Coefficient, Including Non-Informative and Unassigned Data

We explored potential associations between clinical data and ICSI cycle outcome by calculating Spearman’s rank correlation coefficient and analyzing the two data sets. The first set encompassed all data, including non-informative results, and samples where the result was nil with regard to niPGT-A analysis, while the second data set included only data for embryos whose ni-PGT-A profile was classified as ‘euploid’, ‘aneuploid’, or ‘mosaic’. While the analysis yielded comparable results for both types of data sets, the results presented herein pertain exclusively to the first data set, which is presented in detail in Appendix A. The detailed results of the second data analysis can be found in Appendix A.

Briefly, female age was positively correlated to male age (*p* < 0.000, ρ =0.712), female (*p* = 0.012, ρ = 0.143) and male BMI (*p* = 0.008, ρ= 0.150), and to a level of hormone TSH (*p* = 0.006, ρ= 0.166), while it was negatively correlated with the level of AMH (*p* = 0.003, ρ= −0.201), with the number of blastocysts per cycle (*p* = 0.019, ρ = −0.205), and with embryo quality (*p* = 0.004, ρ= −0.162). On the other hand, male age was positively correlated with female BMI group (*p* = 0.037, ρ = 0.119), with male BMI (*p* < 0.000, ρ = −0.223), with sperm morphology (*p* < 0.000, ρ = 0.392), with the day of SECM sampling (*p* = 0.003, ρ = 0.169), embryo ploidy status (*p* = 0.024, ρ = 0.127), and negatively correlated with embryo quality (*p* < 0.000, ρ = −0.205). There was no significant correlation between male age and pregnancy outcome (*p* = 0.331).

There were several significant correlations between female hormone levels. AMH was significantly positively correlated to the number of retrieved oocytes (*p* < 0.000, ρ = 0.634), embryos (*p* < 0.000, ρ = 0.465) and blastocysts (*p* < 0.000, ρ = 0.364) per cycle, while the number of retrieved oocytes was significantly negatively correlated to FSH (*p* = 0.010, ρ = −0.237). The female BMI was negatively correlated with the hormones FSH (*p* = 0.021, ρ = −0.138), LH (*p* < 0.000, ρ = −0.296), and AMH (*p* = 0.045, ρ = −0.135). More importantly, there was a significantly negative correlation between cfDNA concentration and the hormones FSH (*p* = 0.029, ρ = −0.136) and LH (*p* = 0.047, ρ = −0.124). The only two hormones that had a significant correlation with the day the SECM was collected were AMH (*p* = 0.019, ρ = 0.159) and LH (*p* = 0.046, ρ = 0.121).

We also correlated male age, male BMI and general sperm characteristics with ICSI cycle outcomes (see Appendix A for more detail). Interestingly, there was a significantly negative correlation between male BMI and pregnancy outcome (*p* = 0.003, ρ = −0.336), but there was no statistical significance linking female BMI and pregnancy outcome (*p* = 0.250).

Regarding the measured cfDNA concentration in SECM samples, it was significantly negatively correlated to FSH (*p* = 0.029, ρ = −0.136) and LH (*p* = 0.047, ρ = −0.124) levels.

We found no correlation between cfDNA concentration and female BMI (*p* = 0.785), but we did find a significantly negative correlation between cfDNA and the male BMI (*p* < 0.000, ρ = −0.239). Importantly, there was a significant positive correlation between cfDNA concentration, the day the SECM was sampled (*p* = 0.014, ρ = 0.144), and pregnancy outcome (*p* = 0.020, ρ = 0.272), but there was no correlation to embryo quality (*p* = 0.121) or male age (*p* = 0.530).

## 4. Discussion

In this study we explored the potential of niPGT for the determination of embryonic ploidy status in our IVF setting. In addition, we examined associations between embryonic ploidy status, basic patient characteristics of the infertile couple, and pregnancy outcomes. Our data illustrated that niPGT-A is a feasible approach. However, the risk of maternal DNA contamination remains significant, highlighting the necessity of embryo culture protocol being precisely optimized and validated to minimize this issue. Despite this limitation, the correlation of embryonic ploidy to the basic patient characteristics of the infertile couple reveals many relevant correlations, with the most interesting one being the influence of not just the female age, but of male age and BMI as well. These results indicate the need for further research into how general patient characteristics affect embryo development, especially those factors that may be modifiable. This study highlights the importance of both partners’ age and physical health, and how these parameters contribute to the health and early embryogenesis of the embryo.

We used niPGT-A as the primary method of analyzing embryonic genetic material. Because of the legislation in our country, we were restricted from using invasive PGT-A as an additional method of analyzing and confirming embryo ploidy. The only available method to assess the accuracy of the results was to track births and indirectly confirm the sex and ploidy status of the fetus based on the genetic health and sex of the offspring. In our study we observed a high level of maternal contamination, which we were able to only confirm indirectly. Specifically, we observed a discrepancy in 59% of cases where males were born despite the genetic analysis indicating a “euploid, female” result. This directly confirms maternal contamination, as we used the ICSI method and paternal contamination is in any way not feasible. The article by Barzanouni et al. also reported maternal contamination when trying to predict the sex of the embryo [34]. In their study they diagnosed the X chromosome via the FMR1 gene, and the Y chromosome via the SRY gene. They proposed that the low amount of cfDNA extracted from SECM was responsible for their results. In their study, they determined the sex of 30 embryos on day 5. The genetic results showed that 7 out of 30 embryos were male (23.3% of embryos), but when they conducted an ultrasound screening, 13 out of 30 embryos were male (43.3% of embryos). They estimated the sensitivity of their method was approximately 53.84% while the negative predictive value was approximately 73.9% [34]. There have been other notable studies in the last decade that describe the plausibility of maternal contamination and question the suitability of the niPGT-A method. In the study by Vera-Rodriguez et al. they aimed to better understand cfDNA composition [19]. They studied the cfDNA from embryos fertilized by the ICSI method and compared it to DNA collected via TE biopsy of those same embryos. They observed a discordance between the cfDNA results and the results from the TE biopsy in approximately 67% of the cases. The difference was mainly attributed to maternal contamination in the SECM, which was proven by analyzing single-nucleotide polymorphisms in DNA from SECM, TE biopsy, and follicular fluid samples to detect maternal contamination [19]. Feichtinger et al. analyzed the spent culture of oocytes with fertilization failure and reported ‘notable maternal contamination’ [35]. A study by Chen et al. in 2021 studied DNA methylome in SECM [36]. By identifying specific differentially methylated regions, they were able to attribute the DNA to blastocysts, cumulus cells, or polar bodies. They observed that the cumulus cells contributed to more severe contamination than the polar bodies. The cumulus contaminated SECM in 50% of the samples, and of these approximately half were moderately contaminated and approximately half were severely contaminated. Meanwhile, polar bodies contaminated 27% of SECM samples, of which most of the samples were moderately contaminated. [36]. In the extensive review by Hammond et al., they proposed that maternal contamination originated from maternal cumulus cells [37].

Despite the indirect confirmation of maternal contamination, we nevertheless observed a significant decrease in embryo ploidy rate between the oldest and the youngest group and an increase in the rate of embryos classified as mosaic. This indicates that female age plays an important role in embryo ploidy, as proposed before [38,39]. We also observed a significant correlation between female age, but not female BMI, and embryo ploidy status. More precisely, the older the women, the more likely it is for the embryo to be aneuploid or mosaic. This is important, as it has been the conclusion of several prominent articles in the recent history of ART articles indicating that age correlates with embryo aneuploidy [40,41] or the complexity of mosaicism (which could lead to failed correction mechanisms and the prevail of aneuploid cell lines) [42,43,44].

In the recent years, the embryonic ability to self-correct has been of great interest to the scientific community [45,46,47]. In our study, we observed a significant correlation between cfDNA concentration and embryo ploidy. Our findings revealed that the higher the cfDNA concentration, the higher the chances of the embryo being aneuploid or mosaic. This observation could indirectly concur with the idea of embryonic self-correction ability as previously proposed by Orvieto et al. [45] and later by Wang et al. [47].

When correlating basic characteristics of the infertile couple with embryo ploidy status and other ICSI cycle outcomes, several interesting significant correlations emerged. Notably, we observed a correlation between cfDNA concentration in SECM and embryo ploidy. Of particular interest, however, was the fact that we observed a significant positive correlation between cfDNA concentration and male BMI (but not female BMI). In our view, the most compelling finding was the significant positive correlation between male BMI and cfDNA concentration—a relationship not previously reported in the literature to our knowledge. While cfDNA origin in SECM has not yet been confirmed, it is speculated that it is derived from the blastocyst, cumulus cells, and polar bodies [36]. Chen et al. observed that nearly half of SECM samples were contaminated with genetic material from cumulus cells, and one third was contaminated with genetic material from polar bodies [36]. They concluded that net maternal contamination in their experiment was greater than 60% in approximately one third of the samples. They also concluded that only a third of the samples had a net maternal contamination of less than 20% [36]. Considering their results of net maternal contamination, we are unable to explain the elevated cfDNA concentration in correlation to male BMI or, in fact, any male characteristic. Approaching this issue from a different perspective may offer further insight. A 2012 study by Colaci et al. described that the odds of live birth among couples undergoing ICSI to be 84% lower in couples with obese male partners, compared to couples with men with normal BMI, though they did not attempt to explain the phenomenon [9]. In our study, which exclusively used the ICSI method, we similarly observed a significant negative correlation between male BMI and pregnancy outcome. To explore this association further, it is important to consider recent studies examining embryo ploidy in relation to male BMI and sperm parameters. Notably, a 2020 study by Wang et al. found a negative correlation between male BMI and an increased risk of chromosomal aberration-related miscarriages [48]. In the study by Yang et al., they observed a correlation between abnormal semen parameters and an increase in embryo aneuploidy rate [49]. It has long been hypothesized that in early embryogenesis, embryos are capable of self-correction in an effort to eliminate the aneuploid cell lineage [50]. A 2020 study proposed this could be achieved through an autophagy-mediated cell apoptosis [51]. This could, in turn, elevate cfDNA concentrations in SECM. Furthermore, our study successfully correlated male BMI and sperm characteristics, both of which could dependently or independently have contributed to embryo ploidy.

More specifically, we observed a correlation between male BMI and sperm morphology and motility, even though no such correlation was observed between male BMI and sperm concentration or volume. There are other studies, published in the past, showing that sperm quality and integrity are in a close relationship to male BMI. These studies, however, have offered various conclusions [52,53,54].

All of these studies indicate that sperm quality and integrity are in close relationship with male BMI. It is important to acknowledge that numerous other intrinsic and extrinsic factors may contribute to sperm quality and integrity. Among these are macro and trace element concentrations, many of which are present in the environment and are known regulators of biological processes, including reproduction [55].

Furthermore, within our cohort, a significant correlation was observed between male BMI and male age, indicating that older male participants had a higher BMI.

When observing individual parameters, male age in our study correlated to embryo quality category. Specifically, increased male age was correlated with a higher likelihood of embryos being aneuploid or mosaic. This observation stands in contrast with several studies that found no correlations between advanced paternal age and aneuploidy [56,57,58].

One of the limitations of our study was undoubtedly the standardized laboratory procedure in culturing embryos. To better reflect a clinical setting, we designed our study in a way to adhere to the standardized procedures of our IVF Laboratory. We started our study in the year 2021, when there was limited literature available to support the methodology and a necessity for a two-step embryo denudation and cultivation. Therefore, we used a single-step procedure in which embryos were cultured for 5/6 day in the same medium without medium changes, which probably led to maternal DNA contamination. Another limitation in our study was the small sample size of couples per age group. While we did include 131 couples in our study, the number of couples per age group varied between 29–38 couples. Nevertheless, we included 77 embryos in Group 1, 105 embryos in Group 2, 73 embryos in Group 3, and 62 embryos in Group 4. Therefore, further studies with revised protocols and a larger number of included couples per age group are needed. Another limitation of this study is the limitation of sequencing depth when analyzing cfDNA. The usual industry recommendation is 500,000 single-end reads using 100 bp and 1,000,000 base bin size. However, the recommendation for detection of very low-grade mosaicism and segmental aneuploidy the number of single-end reads is higher, at around 2,000,000 reads and an even lower bin size. We used a commercial service that does not report mosaicism below 50%, as the technique and the clinical implications are limited in cases of low-grade mosaicism for the aforementioned reasons. We speculate that this could be avoided in the future with further development of the analyzing technique. An important limitation was the time frame of our research. The patients were included prospectively, as were their characteristics and SECM samples of the embryos. The analysis itself, however, was done retrospectively. This resulted in a limited amount of information on patient characteristics, as it was impossible to determine or predict certain parameters for the past. We speculate that this issue could be mitigated through the implementation of a comprehensive questionnaire on each couple’s health that should be provided upon entry into the study.

## 5. Conclusions

This study is distinctive in several aspects. While it indirectly highlights the importance of standardized protocols in embryo cultivation and denudation prior to niPGT-A through encountered maternal contamination, it also offers valuable insights into the impact of both maternal and paternal health on early embryogenesis. We indirectly proved the presence of embryonic self-correcting mechanisms by measuring cfDNA in SECM. This is also one of the first studies that observed elevated embryonic cfDNA concentrations in SECM in relations to basic paternal parameters, showcasing the importance of not only maternal but also paternal health prior to conception.

Through this study, we have broadened the scientific lens through which infertility is viewed, incorporating male infertility into the broader scope of scientific exploration.

## Figures and Tables

**Table 1 biomolecules-15-01215-t001:** Demographic, endocrine, and clinical characteristics of women and men included into this study. Values are reported as median with interquartile range (Q1–Q3). *p*-value under 0.05 was recognized as statistically significant.

	Group Categories				
	Group 1	Group 2	Group 3	Group 4	*p*-Value
Number of couples	29	38	33	31	
Female age (years)	27.0 (26.0–28.5)	32.6 (31.4–33.6)	37.4 (36.1–38.9)	41.5 (40.9–42.4)	<0.001 (1 vs. 2 *p* = 0.002, 3 vs. 4 *p* = 0.001, all other comparisons *p* < 0.001)
Male age (years)	31.0 (28.1–33.1)	34.0 (32.0–38.7)	39.7 (36.9–44.0)	43.9 (40.7–47.4)	<0.001 (1 vs. 2 *p* = 0.210, 1 vs. 3 *p* < 0.001, 1 vs. 4 <0.001, 2 vs. 3 *p* = 0.002, 2 vs. 4 *p* < 0.001, 3 vs. 4 *p* = 0.174)
Female FSH [IU/L]	7.6 (6.0–8.7)	7.4 (6.1–10.1)	7.2 (6.2–8.9)	7.7 (6.9–10.2)	0.720
Female LH [IU/L]	4.6 (3.4–5.8)	5.1 (3.5–7.8)	4.6 (3.8–6.5)	5.2 (3.4–6.9)	0.806
Female AMH [µg/L]	2.8 (2.2–4.8)	2.8 (1.2–5.6)	2.9 (1.0–3.8)	1.4 (0.8–2.6)	0.031 (1 vs. 4 *p* = 0.034, 2 vs. 4 *p* = 0.174, 3 vs. 4 *p* = 0.323, all other comparisons *p* = 1)
Female PRL [µg/L]	10.0 (6.7–15.5)	8.7 (5.8–12.0)	9.9 (6.8–14.5)	9.0 (6.9–12.1)	0.433
Female TSH [mIU/L]	1.9 (1.2–2.5)	1.9 (1.2–2.3)	2.1 (1.3–2.7)	2.1 (1.5–2.9)	0.252
Female BMI [kg/m^2^]	24.2 (21.8–30.3)	23.5 (21.2–25.8)	23.9 (20.9–27.4)	26.0 (21.9–28.4)	0.372
Male BMI [kg/m^2^]	26.5 (24.6–30.9)	26.4 (24.0–29.6)	26.0 (24.0–28.6)	26.3 (24.7–29.8)	0.836
Total dose of used Gonadotrophins in IE	1525 (1350–2193.8)	2025 (1500–2475)	2250 (1800–3037.5)	2475 (2025–2925)	*p* < 0.001 (1 vs. 2 *p* = 0.251, 1 vs. 3 *p* = 0.004, 1 vs. 4 *p* < 0.001, 2 vs. 4 *p* = 0.152, 2 vs. 3 *p* = 0.705, 3 vs. 4 *p* = 1)

**Table 2 biomolecules-15-01215-t002:** Characteristics of ICSI cycle outcomes according to Age Group Categories. Values are reported as median with interquartile range (Q1–Q3) or proportions. A *p*-value under 0.05 was recognized as statistically significant.

	Age Group Categories				
	Group 1	Group 2	Group 3	Group 4	*p*-Value
Number of retrieved oocytes; *n* (per cycle)	258 (8 (4.5–11.5))	396 (9.0 (5.0–12.3))	280 (8.0 (4.5–11.0)	243 (8.0 (4.0–11.0))	0.591
Number of normally fertilized oocytes per number of oocytes injected	116 (59.2%)	192 (64.2%)	132 (62.3%)	119 (63.0%)	0.727
Immature oocytes; *n* (rate (%))	47 (18.6)	74 (18.7%)	46 (16.4%)	41 (16.9%)	0.864
Degenerated oocytes per number of retrieved oocytes; *n* (rate (%))	36 (14.0%)	47 (11.9%)	34 (12.1%)	32 (13.2%)	0.864
Polyploidies per number of retrieved oocytes; *n* (rate (%))	5 (1.9%)	8 (2.0%)	6 (2.1%)	5 (2.1%)	0.999
Cleaved embryos; *n* (% per zygotes)	114 (98.3%)	192 (100%)	128 (97.7%)	119 (100%)	
Number of embryos per cycle	3.0 (2.0–5.5)	4.5 (2.0–6.0)	3.0 (2.0–5.5)	3.0 (2.0–6.0)	0.357
Number of blastocysts per cycle	2.0 (1.0–3.5)	2.0 (1.0–3.0)	1.0 (1.0–3.0)	1.0 (0.0–2.0)	0.138
Blastocysts per embryos cultured until day 5/6; *n* (rate (%))	79 (69.3%)	103 (53.6%)	70 (51.6%)	55 (46.2%)	0.004 (1 vs. 2 *p* = 0.007, 1 vs. 3 *p* = 0.020, 1 vs. 4 *p* < 0.001, 2 vs. 3 *p* = 0.857, 2 vs. 4 *p* = 0.204, 3 vs. 4 *p* = 0.184)
Good quality blastocyst per total number of blastocysts (rate (%))	52 (65.8%)	67 (65.0%)	48 (68.6%)	36 (65.5%)	0.968
Fair quality blastocyst per total number of blastocysts (rate (%))	18 (22.8%)	18 (17.5%)	14 (20.0%)	11 (20.0%)	0.851
Poor quality blastocyst per total number of blastocysts (rate (%))	9 (11.4%)	18 (17.5%)	8 (11.4%)	8 (14.5%)	0.598

**Table 3 biomolecules-15-01215-t003:** Embryo utilization and pregnancy outcomes based on age group category.

	Age Group Categories				
	Group 1	Group 2	Group 3	Group 4	*p*-Value
Embryo utilization (transferred + frozen embryos); *n* (rate (%))	77 (67.5%)	100 (52.1%)	77 (60.2%)	63 (52.9%)	0.039 (1 vs. 2 *p* = 0.008, 1 vs. 3 *p* = 0.234, 1 vs. 4 *p* = 0.023, 2 vs. 3 *p* = 0.156, 2 vs. 4 *p* = 0.880, 3 vs. 4 *p* = 0.254)
Cryopreserved blastocysts; *n* (rate (% of all embryos))	52 (45.6%)	74 (38.5%)	47 (36.7%)	31 (26.1%)	0.019 (1 vs. 2 *p* = 0.222, 1 vs. 3 *p* = 0.162, 1 vs. 4 *p* = 0.002, 2 vs. 3 *p* = 0.741, 2 vs. 4 *p* = 0.024, 3 vs. 4 *p* = 0.072)
Cycles with at least one blastocyst on day 5; *n* (%)	26 (89.7%)	34 (89.5%)	28 (84.8%)	22 (71.0%)	0.141
Cycles with embryo cryopreservation; *n* (%)	15 (51.7%)	22 (57.9%)	19 (57.6%)	11 (35.5%)	0.232
Cycles with freezing without ET; *n* (%)	4 (13.8%)	7 (18.4%)	4 (12.1%)	2 (6.5%)	0.531
Cycles without freezing/without ET; *n* (%)	0	0	0	0	
Total number of fresh ETs	25	31	29	29	
Pregnancies; *n* (% per fresh ET)	14 (56.0%)	13 (41.9%)	11 (37.9%)	10 (34.5%)	0.411
Pregnancies; *n* (% per frozen ET)	7 (35.0%)	12 (35.3%)	3 (12.0%)	2 (18.2%)	0.160
Cumulative pregnancy rate; *n* (% per number of all couples)	21 (72.4%)	23 (60.5%)	14 (42.4%)	11 (35.5%)	0.015 (1 vs. 2 *p* = 0.308, 1 vs. 3 *p* = 0.017, 1 vs. 4 *p* = 0.004, 2 vs. 3 *p* = 0.129, 2 vs. 4 *p* = 0.038, 3 vs. 4 *p* = 0.569)
Cumulative delivery rate; *n* (% per number of all couples)	13 (44.8%)	18 (47.4%)	8 (+1 ongoing) (27.3%)	5 (16.1%)	0.023 (1 vs. 2 *p* = 0.834, 1 vs. 3 *p* = 0.087, 1 vs. 4 *p* = 0.016, 2 vs. 3 *p* = 0.043, 2 vs. 4 *p* = 0.006, 3 vs. 4 *p* = 0.418)
Couples that didn’t get pregnant/deliver, but still have frozen embryos	0/3	2/3	4/5	0/2	

**Table 4 biomolecules-15-01215-t004:** Collection of samples and cfDNA analysis in different age groups.

	Age Group Categories				
	Group 1	Group 2	Group 3	Group 4	*p*-Value
Spent medium samples collected on day 5	72 (93.5%)	86 (86.0%)	68 (88.3%)	54 (85.7%)	0.400
Spent medium samples collected on day 6	5 (6.5%)	14 (14.0%)	9 (11.7%)	9 (14.3%)	0.400
DNA collected (Library Quantification [ng/µL]) day 5	9.82 (4.86–21.98)	14.0 (6.39–24.2)	14.45 (9.02–22.08)	11.95 (6.84–19.3)	0.166
DNA collected (Library Quantification [ng/µL]) day 6	13.2 (10.72–28.4)	18.6 (11.85–38.25)	23.6 (10.77–36.9)	11.7 (9.72–22.75)	0.339
DNA collected (Library Quantification [ng/µL]) all samples	10.04 (4.95–22.53)	15.7 (7.08–28.05)	14.7 (9.03–23.05)	11.8 (7.46–19.3)	0.055
*p*-value (comparison for amount of DNA collected between day 5 and 6)	0.203	0.088	0.191	0.691	
Euploid embryo rate (%)	66 (85.7%)	79 (79.0%)	63 (81.8%)	42 (66.7%)	0.042 (1 vs. 2 *p* = 0.250, 1 vs. 3 *p* = 0.509, 1 vs. 4 *p* = 0.008, 2 vs. 3 *p* = 0.638, 2 vs. 4 *p* = 0.080, 3 vs. 4 *p* = 0.039)
Male euploid embryo rate	6 (9.1%)	4 (5.1%)	3 (4.8%)	4 (9.5%)	0.609
Female euploid embryo rate	60 (90.9.%)	75 (94.9%)	60 (95.2%)	38 (90.5%)	0.609
Mosaic embryo rate (%)	2 (2.6%)	0	3 (3.9%)	6 (9.5%)	0.043 (1 vs. 2 *p* = 0.105, 1 vs. 3 *p* = 0.653, 1 vs. 4 *p* = 0.078, 2 vs. 3 *p* = 0.047, 2 vs. 4 *p* = 0.002, 3 vs. 4 *p* = 0.177)
Aneuploid embryo rate (%)	6 (7.8%)	9 (9.0%)	7 (9.1%)	10 (15.9%)	0.407
Unassigned data, Non-informative, without diagnosis	3 (3.9%)	12 (12.0%)	4 (5.2%)	5 (7.9%)	0.178

## Data Availability

The original contributions presented in this study are included in the article/Appendix A. Further inquiries can be directed to the corresponding author(s).

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
