# Peer review of "The Role of Preconception Parental Health on Embryo Quality—Preliminary Results of a Prospective Study Using Non-Invasive Preimplantation Genetic Testing for Aneuploidy"

_biomolecules, 2025, doi:10.3390/biom15091215_

Round 1

Reviewer 1 Report (Previous Reviewer 2)

Comments and Suggestions for Authors The revised manuscript has incorporated a post hoc power analysis using the G*Power program, revealing a statistical power of 0.64. Based on the calculated effect size (w), the authors estimated that approximately 100 additional embryos would be needed to achieve adequate power (0.80). Although a power of 0.64 is considered moderate to low and implies an increased risk of Type II error, the topic is novel and provides valuable preliminary insights. The manuscript may be considered for publication as an exploratory study that could inspire further research in this area.

Author Response

Thank you for your conformation and time.

Kind regards

Reviewer 2 Report (Previous Reviewer 3)

Comments and Suggestions for Authors

I think that the manuscript has been improved and now is acceptable in the present form

Author Response

Thank you for your conformation and time.

Kind regards

Reviewer 3 Report (New Reviewer)

Comments and Suggestions for Authors

This study enrolled 131 infertile couples undergoing assisted reproductive treatment and collected 316 spent embryo culture medium samples. Participants were stratified into four groups based on female age. Using non-invasive preimplantation genetic testing for aneuploidy (niPGT-A), the study analyzed cell-free DNA (cfDNA) in the culture medium to assess embryonic ploidy status. The authors systematically compared parental health characteristics—including age, body mass index (BMI), and hormonal profiles—with the incidence of embryonic aneuploidy. The results confirmed a strong association between advanced maternal age and increased aneuploidy rates, as well as reduced pregnancy outcomes. Notably, the study also revealed a significant correlation between male BMI and both cfDNA concentration and pregnancy outcomes, highlighting a previously underrecognized role of paternal health in embryonic genetic integrity.

While this manuscript presents data from a prospective study utilizing non-invasive preimplantation genetic testing for aneuploidy (niPGT-A) and explores associations between parental health and embryonic ploidy, I find that the study’s novelty is limited.

The authors suggest a potential link between male health and embryo aneuploidy, but the only male parameter assessed is BMI. Although this observation could imply the role of sperm quality in embryonic ploidy, the study lacks direct clinical semen parameters (e.g., morphology, motility, DNA fragmentation index), which limits the strength and interpretability of this conclusion.

Furthermore, the claim that male BMI is associated with cfDNA concentration and pregnancy outcomes, while interesting, does not substantially advance the field without supporting mechanistic data or broader and more detailed male reproductive metrics. Thus, I strongly suggest that the authors provide more data on male health metrics, especially clinical semen parameters.

The statistical analysis section is also insufficiently described. The authors mention the use of the Kruskal-Wallis test for independent samples in the event of non-normal distribution, but they do not clarify the specific statistical tests applied to each dataset. In particular, the comparison of cumulative pregnancy rates using pairwise methods is not adequately justified and may not be appropriate in this context. Each table should clearly specify which statistical test was applied, along with any post-hoc methods used.

Minor comments:

The manuscript mentions “Table 5,” but I was unable to find this table in the current version. Please verify the table numbering and ensure that all tables are correctly labeled and included in the document.

Some sentences are unclear. For example, the sentence “Furthermore, there was no correlation with female BMI, but we did find a significantly negative correction with the male BMI” is unclear. Typographical errors should be corrected, and the text should be reviewed carefully.

The prevalence of maternal DNA contamination in this study should be compared to contamination rates reported in other high-quality studies. Such a comparison would be valuable and help the reader assess the reliability of the results in the current study.

Author Response

Thank you for your time.

Our response is added in the attached Word document.

Kind regards

Reviewer 4 Report (New Reviewer)

Comments and Suggestions for Authors

In this article, the authors analyzed aneuploidy and mosaicism in cfDNA in SECM. They have collected oocytes from 4 groups of women in different age groups. They also collected clinical data from the mother and the father. They have observed a significant negative correlation between male body mass index and cfDNA concentration in SECM. They also observed that male age was positively correlated with embryo ploidy status and negatively correlated with embryo quality. Also, male BMI was negatively correlated with pregnancy outcome. 

The study and findings are interesting. However, I do have some concerns about how the data was analyzed and the basic premise of the study. As the authors acknowledge in the manuscript, cfDNA niPGT-A is not a standard method to determine the embryo ploidy for PGT. The basic premise of the study is to utilize cfDNA-based non-invasive PGT-A to investigate the embryo quality associated with the parental health. 

The cfDNA itself has high contamination of maternal DNA and possible contamination of a mosaic embryo. The concordance studies that this study cited show 30-70% matching aneuploidy patterns between TE biopsy and cfDNA. That is very concerning, and I don’t know whether that is acceptable FDR for any study because this study refers to the number strictly correlates to the health of the embryo. The power calculation would have also adjusted based on this. In the manuscript, the authors acknowledge that. 

According to Ver-Rodriguez et al, 2018, only 8% of DNA is from embryonic DNA, with 75% of the analyzed chromosomes being concordant with the trophectoderm DNA. If only 8% of DNA is from embryo, the sequencing depth should have been a lot higher. They mentioned that their sequencing depth was around 75 Mb (500,000 single-end reads with 150 bp = 75 Mb) with 10 Mb segments. As it is, it will not be enough to report mosaicism, and if only 8% of the DNA from the embryo, it will not be able to detect aneuploidy.  

Moreover, in this study, embryos were cultured for 5 or 6 days without media change, which likely increased the contamination from maternal DNA and mosaicism. The manuscript also mentioned that they did not filter out the maternal DNA contamination. There could have been different methods that they could have done pre (fragment size selection) and post (SNP genotyping and allele frequency analysis). Failure to do so would have increased the number of euploidy samples. 

One other critical problem I also observe is that some of the numbers in Table 4 do not match the numbers from other tables. For example, the number of blastocytes was 79, 103, 70, and 55, but the SECM used in the NGS study is 77, 100, 77, and 63. I tried to match the embryo utilization number, but the number is still greater for groups 3 and 4. If they utilized more unhealthy embryos in groups 3 and 4, it would most likely skew the cfDNA concentration. It needs to be explained further in the method.

The discussion needs to cover more of the limitations of the study. 

Author Response

Thank you for your time.

Our response is added in the attached Word document.

Kind regards

This manuscript is a resubmission of an earlier submission. The following is a list of the peer review reports and author responses from that submission.

Round 1

Reviewer 1 Report

Comments and Suggestions for Authors

I believe the manuscript combines multiple concepts, making the reading cumbersome. On the one hand, it attempts to justify the use of ni-PGT-A, even though this is not the main objective of the study. On the other hand, the findings related to paternal "health" are not consistent.

Furthermore, the number of analyzed blastocysts is limited, making the results not generalizable. It would have been important to highlight the study’s limitations more clearly.

Acronyms should not be used in the abstract (e.g., SECM).

There are numerous grammatical errors, including some syntactical issues.

Results

Table 1 is unnecessary; a textual description of the groups would suffice.

The description of Table 2 is incorrect. For instance, it inaccurately refers to significant differences in AMH or FSH values.

Discussion

The discussion is excessively long and difficult to follow.

The initial objective of the study is overshadowed by secondary and non-substantial findings.

Comments on the Quality of English Language

There are numerous grammatical errors, including some syntactical issues.

Reviewer 2 Report

Comments and Suggestions for Authors

The manuscript titled “The Role of Preconception Parental Health on Embryo Quality; A Prospective Study Using Non-invasive Preimplantation Genetic Testing for Aneuploidy” retrospectively evaluates the association between embryonic ploidy status assessed by noninvasive preimplantation genetic testing for aneuploidy and basic patient characteristics of the infertile couple. The authors report a significant correlation between male BMI and cfDNA concentration in SECM, as well as between male BMI and pregnancy outcome. Notably, the link between male BMI and cfDNA concentration appears independent of maternal factors. This study provides a novel perspective on the paternal influence in embryonic ploidy status.

As an exploratory and innovative study, it brings a valuable contribution to the field. However, the manuscript does have some limitations including those related to embryo culture procedures and the relatively small sample size. Despite these, the findings are of interest and merit publication in Biomolecules, provided minor revisions are addressed.

Minor revisions:

  • Sample Size and Statistical Analysis: The study mentions statistical significance in several instances, but the relatively small sample size in certain groups (e.g., 29–38 couples per age group) may limit the generalizability of the findings. It would be useful to include a more robust discussion on the power analysis used to determine the sample size and address potential limitations due to the sample size in the results section.
  • Please list all abbreviations used in abstract (e.g., SECM, NGS, cfDNA) to improve clarity for readers unfamiliar with the terminology.
  • In Table 5, the parentheses are missing for the item “Euploid embryo rate (%)” on line 9 and list 1-please correct this formatting issue.

Reviewer 3 Report

Comments and Suggestions for Authors

In the article titled: “The Role of Preconception Parental Health on Embryo Quality;  A Prospective Study Using Non-invasive Preimplantation  Genetic Testing for Aneuploidy” the authors  aimed to correlate embryonic ploidy status studied with non-invasive preimplantation genetic testing for aneuploidy with basic patient characteristics of the infertile couple, to gain insight on the effects of parental physical health on embryo ploidy.

I find an interesting work but some information is missing. Therefore, I suggest a major revision.

 The suggestions I would make are as follows:

  • µl must be corrected in µL
  • Enrich the materials and methods with details, add among other things the criteria for selecting subjects by including lifestyles, drinkers? smokers? drug users? contact with toxic substances in the workplace? physical activity?
  • The initial part of the results in my opinion should go into the materials and methods. I find the materials and methods poor and unclear
  • There are many causes that can lead to DNA fragmentation, male infertility. i believe that not only the BMI of the male partner should be taken into consideration.
  • Among the causes of male infertility is environmental pollution. In the work I would also talk about that. In this regard, I recommend reading and citing the following works 10.3390/ijerph191811635; 10.3390/ijerph191711023. Any papers recommended in the report are for reference only. They are not mandatory. You may cite and reference other papers related to this topic.
  • Better define the limitations of this study.
  • The paper needs of english check
Comments on the Quality of English Language

The English could be improved to more clearly express the research

Round 2

Reviewer 3 Report

Comments and Suggestions for Authors

The authors addressed all my questions. I accept the manuscript in the present form